# OpenReview forum: "IF-Guide: Influence Function-Guided Detoxification of LLMs"
_NeurIPS.cc/2025/Conference — NeurIPS 2025 poster_

### Official Review · Reviewer_AngH · 2025-06-30

**Clarity:** 3
**Significance:** 3
**Originality:** 2
**Rating:** 4
**Confidence:** 4

**Summary:**

This work studies how to detoxify LLMs.  The work first examines the normal influence function and demonstrates that it will impact the model’s utility by misclassifying nontoxic examples.  Then the authors propose a new token-level influence function along with a new loss function.  Training with If-guide can significantly reduce the toxicity of LLMs in comparison with commonly used filtering methods and even outperform DPO and RAD in toxicity. The authors also show that models trained with If-guide are more robust to gcg attack.

**Questions:**

1 It may be better if the reasons why the normal influence functions do not work are clearer. For example, comparing the Influence function with detoxify in detecting toxic examples.

2 what does the DPO and RAD in table 1 refer to? Do they mean, e.g., DPO is applied to the base model trained without any filtering? A cleaner comparison should be IF-Guide, word/ toxicity filtering, and then DPO/RAD applied to those respective pre-trained models.  Current Table 1 is a bit confusing.

3 instead of dropping examples, token-level attribution is proposed to improve the utility and retain benign knowledge. But If-guide is somehow always worse than direct training with filtering on the utility task. Can you provide some explanation or hypothesis? Is If-guide better than using naive influence function in the utility tasks?

4 On the other hand,  toxicity is reduced more than by filtering methods or the conventional influence function. But the motivation of if-guide is to avoid degradation in utility, as you stated in section 3.2.  Could you provide some explanations? Do you try example filtering with your newly designed influence function? is the accuracy better than word filtering or detoxify?  Does suppressing toxic tokens instead of dropping them improve utility?


Others:

The legend of figure  5 seems wrong.

Table 1 is not consistent. The `` word filtering'' is sometimes at different row.

**Ethical Concerns:**

["NO or VERY MINOR ethics concerns only"]

**Final Justification:**

Thanks for clarification. I hope the author could make corresponding justifications, clarifications and updates in their next revision. I will maintain my score.

**Limitations:**

yes

**Quality:**

2

**Strengths And Weaknesses:**

Strength

the work demonstrates the limits of influence functions in detoxifying

The proposed If-guide has impressive performance in detoxifying

The proposed If- guide can be applied during pre-training or post-finetuning where the method is always effective in detoxifying

Weakness:

If-guide may reduce the utility as a trade-off.  Although the authors design different components to avoid that, if-guide tends to given worse performance than other methods on influency.

The experiments do not clearly showcase the method's contributions. If-guide mainly consists of (1) a new influence function (2) a new loss function for training, which are proposed to address the performance degradation of using a naive influence function. Because naive influence function may return nontoxic but important examples.  However, the results show that **IF-Guide** is significantly more effective at detoxification compared to the baselines, rather than maintaining the utility. This makes it unclear what contributions those designed components truly bring. For example, toxificity filtering can be combined with proposed loss function. If this method is worse than If-Guide, it suggests the influence function indeed identify some really important tokens. Overall, despite the impressive performance, more discussion on explaining why the method works may be more helpful and interesting.

The overall framework of If-guide is very close to this related work [1] apart from the influence function part. This makes it more important to explain the role of influence function which is lacking in the current version.

[1] Korbak, Tomasz, et al. "Pretraining language models with human preferences." *International Conference on Machine Learning*. PMLR, 2023.

---

> ### Author Rebuttal · Authors · 2025-07-31
>
> We thank the reviewer for taking the time to read and evaluate our work. We also appreciate the recognition of its strengths. Below, we address the reviewer’s concerns and questions. We will incorporate this discussion into the final version of our paper for completeness.
>
> ---
>
> **(Weakness) IF-Guide’s Impact on Utility**
>
> We first clarify that, although fluency alone may suggest that IF-Guide has worse performance than the baselines, our method consistently achieves the best overall toxicity–fluency trade-off. To quantify this trade-off, we compare how much accuracy each method sacrifices to reduce a unit of toxicity, using the following metric:
>
> $$\frac{\text{Acc.}}{\text{mean}(\text{EMT}, \text{TP})};$$
>
> We did not use perplexity as it would disproportionately influence the denominator. We report the tradeoff using our results in Table 1. The best trade-off is **bolded**:
>
> |Defense|Pythia-160M|Pythia-410M|Pythia-1B|Llama-3.2-1B|
> |----------------------|:-------------:|:-------------:|:-------------:|:-------------:|
> |**Word Filtering** |1.078|1.094|1.099|1.155|
> |**Toxicity Filtering**|1.369|1.345|1.342|1.373|
> |**DPO**|1.398|1.252|1.251|1.099|
> |**RAD**|4.311| 3.767|3.480|3.453|
> |**IF-Guide (Ours)**|**5.587**|**4.120**|**5.071**|**4.198**|
>
> IF-Guide achieves the highest trade-off across all models, outperforming the baselines by 1.1–5.2$\times$. Given that IF-Guide introduces a small absolute change in fluency (0.93–5.18 for PPL and 0.01–0.06 for Acc.), this trade-off is favorable and highlights the effectiveness of our approach. We will include this comparison in the Appendix of the final version of our paper.
>
> ---
>
> **(Weakness 2) Contribution of IF-Guide’s Components**
>
> We thank the reviewer for this suggestion. To evaluate, we compare IF-Guide to three ablation baselines:
>
> - **Toxicity Filtering + Suppression**: This tests whether off-the-shelf classifiers are sufficient for token selection. We use Detoxify to find the most toxic training sequences and apply our suppression objective to them.
> - **Naive Influence + Suppression**: This tests the value of our attribution techniques over naive influence alone. We apply EK-FAC to select top-scoring tokens, without using our attribution or token selection method.
> - **IF-Guide + Filtering**: This tests the importance of suppression-based training. We disable suppression by setting $\lambda = 0$, filtering toxic tokens instead.
>
> The table below reports toxicity (on RTP) and fluency for each method when applied to Pythia-410M.
>
> |**Method**|**All Prompts**||**Toxic Prompts**||**Non-Toxic Prompts**||**OWT**|**LAMBADA**|
> |------------------------------------------|:-------------:|:-------------:|:-------------:|:-------------:|:-------------:|-------|:-------------:|:-------------:|
> ||EMT|TP|EMT|TP|EMT|TP|PPL|Acc.|
> |**IF-Guide (Ours)**|0.135|0.085|0.184|0.132|0.086|0.037|21.88|0.462|
> |**Toxicity Filtering + Suppression**|0.272|0.211|0.323|0.271|0.221|0.151|88.43|0.169|
> |**Naive Influence + Suppression**|0.445|0.434|0.646|0.670|0.245|0.199|21.04|0.481|
> |**IF-Guide + Filtering**|0.301|0.263|0.426|0.409|0.176|0.117|20.78|0.482|
>
> IF-Guide achieves 2–5.1$\times$ better toxicity reduction than the controlled baselines with similar utility (except for Detoxify; we find that the model selected far too many benign tokens, which substantially degraded utility). These results validate the effectiveness of our proposed contributions and help explain IF-Guide’s effectiveness.
>
> ---
>
> **(Weakness 3) Comparison to Korbak et al.**
>
> While Korbak et al. demonstrate some effectiveness in reducing toxicity, their approach is fundamentally different from ours in many ways:
>
> - **(1) Objective:** Kobak et al. optimize for “general reward alignment” via human preference signals, whereas we target “safety-specific behavior” (toxicity suppression) via “data attribution”.
>
> - **(2) Method:** Their approach uses preference-weighted reward modeling “at the sequence level”, while ours relies on “counterfactual influence estimate” to attribute and suppress toxic behavior “at the token level”.
>
> - **(3) Granularity:** IF-Guide provides fine-grained, interpretable control over model behavior, in contrast to the broader alignment achieved through preference modeling.
>
> These differences highlight the novelty of our contributions. While prior work has explored human preferences to reduce model harms [1, 2, 3], to our knowledge, we are the first to apply influence functions to this problem. Our approach also avoids dependence on expensive and potentially noisy human feedback [4, 5], and enables fine-grained, token-level attribution that can be directly coupled with our novel suppression objective for more effective toxicity reduction.
>
> This positions our work as the first of its kind in this exciting and tangential direction.
>
> [1] Ouyang et al., Training Language Models to Follow Instructions with Human Feedback, NeurIPS (2022)
>
> [2] Rafailov et al., Direct Preference Optimization: Your Language Model is Secretly a Reward Model, NeurIPS (2023)
>
> [3] Ziegler et al., Fine-Tuning Language Models from Human Preferences, arXiv preprint (2019)
>
> [4] Grattafiori et al., The Llama 3 Herd of Models, arXiv preprint (2024)
>
> [5] Casper et al., Open Problems and Fundamental Limitations of Reinforcement Learning from Human Feedback, TMLR (2023)
>
> ---
>
> **(Question 1) Why Do Standard Influence Functions Not Work?**
>
> We identify three fundamental reasons why standard influence functions are ineffective for reducing language model toxicity.
>
> **(1) Document-level scoring is ineffective.** Standard influence functions only provide document-level influence scores. In our preliminary experiments, we attempted to utilize these scores for toxicity reduction; however, we found that suppressing entire documents destabilizes training, and filtering them proved ineffective (as shown in Figure 1). This led us to incorporate token-level attribution.
>
> **(2) Selected tokens lack necessary context.** Even when adding token-level attribution, standard influence functions fail to capture the surrounding toxic context necessary for toxicity reduction. This is affirmed in Appendix D.4 where using a context window size of $w = 0$ reduces the effectiveness of IF-Guide by 2–4$\times$. We thus do not rely entirely on standard influence scores and add additional context.
>
> **(3) High-scoring tokens are not always influential.** It is a known issue in the prior work that standard influence functions often assign high scores to irrelevant tokens [1, 2]. Our preliminary experiments revealed that many top-scoring tokens were erroneous, consisting of many non-toxic articles or repeated characters. We address this limitation with our differential attribution and document-based importance ranking.
>
> [1] Grosse et al., Studying Large Language Model Generalization with Influence Functions, arXiv Preprint (2024)
>
> [2] Choe et al., What is Your Data Worth to GPT? LLM-Scale Data Valuation with Influence Functions, arXiv Preprint (2024)
>
> ---
>
> **(Question 2) DPO and RAD Implementations**
>
> We confirm that the reviewer’s understanding is correct: in Table 1, DPO and RAD are applied to the base model without any filtering. This design choice aligns our evaluation with prior work.
>
> We thank the reviewer for the suggestion. As advised, we applied DPO and RAD to the Word and Toxicity-filtered Pythia-160M models. The table below reports toxicity (on RTP) and fluency for each setting:
>
> | **Method**|**All Prompts**||**Toxic Prompts**||**Non-Toxic Prompts**||**OWT**|**LAMBADA**|
> |--------------------------------|:-------------:|:-------------:|:-------------:|:-------------:|:-------------:|:-------------:|:-------------:|:-------------:|
> ||EMT|TP|EMT|TP|EMT|TP|PPL|Acc.|
> |**Word Filtering + DPO**|0.263|0.228|0.378|0.356|0.148|0.100|26.32|0.471||**Toxicity Filtering + DPO**|0.233|0.194|0.325|0.298|0.141|0.090|26.16|0.461|
> |**IF-Guide (Ours) + DPO**|0.077|0.035|0.101|0.053|0.053|0.017|27.27|0.408|
> |**Word Filtering + RAD**|0.080|0.056|0.131|0.102|0.029|0.009|-|0.438|
> |**Toxicity Filtering + RAD**| 0.067| 0.040 | 0.109| 0.075 | 0.026| 0.004 | -       | 0.444       |
> |**IF-Guide (Ours) + RAD**| 0.031| 0.017 | 0.047| 0.030 | 0.015| 0.004 | -       | 0.438       |
>
> IF-Guide remains the most effective base model. Applying DPO to the IF-Guide base model yields 3–6.5$\times$ greater reductions in EMT and TP compared to applying it to the filtering-based models; with RAD, the improvement ranges from 2.2–3.3$\times$. We will include this experiment in our revision for completeness.
>
> ---
>
> **(Questions 3 and 4) Impact of IF-Guide on Utility and Toxicity**
>
> We acknowledge the ambiguity in our statement in Section 3.2 and appreciate the opportunity to clarify. IF-Guide is designed to *maximize* toxicity reduction while *minimizing* utility degradation—not to eliminate utility loss entirely.
>
> > Do you try example filtering with your newly designed influence function? Is the accuracy better than word filtering or detoxify?
>
> Yes, we show this comparison in Appendix D.4 by setting $\lambda = 0$, which applies filtering instead of suppression. Filtering achieves slightly improved utility—comparable to word and toxicity filtering—but comes at the cost of significantly worse toxicity reduction, performing 2–3$\times$ worse than suppression. This highlights the effectiveness of IF-Guide’s suppression strategy in balancing toxicity reduction with minimal utility degradation.
>
> > Does suppression improve utility compared to dropping tokens?
>
> Suppression incurs a $\sim$5% fluency loss compared to dropping tokens, but achieves 2–3$\times$ more toxicity reduction, resulting in a substantially better overall trade-off. Please refer to our response to Weakness 1 for details.
>
> ---
>
> **Presentation Issues**
>
> We thank the reviewer for pointing these out. We will correct the errors in Table 5 and Figure 5, and carefully review the final version to ensure there are no remaining presentation issues.

---

> > ### Author Response · Authors · 2025-08-05
> > **We Look Forward to Your Feedback**
> >
> > Dear Reviewer AngH,
> >
> > As the deadline for the discussion period is approaching in roughly three and a half days, we would like to inquire whether our response has adequately addressed the questions and concerns you raised.
> >
> > If there are any additional questions, we would greatly appreciate your feedback by tomorrow, so that we have at least 24 hours to respond and address any follow-up concerns.
> >
> > Thank you for your time, and we look forward to hearing from you.
> >
> > Sincerely,
> >
> > The Authors of Paper 25176

---

> > ### Comment · Reviewer_AngH · 2025-08-06
> >
> > Thanks for clarification. I hope the author could make corresponding justifications, clarifications and updates in their next revision.

---

### Official Review · Reviewer_9ufk · 2025-07-02

**Clarity:** 3
**Significance:** 2
**Originality:** 3
**Rating:** 4
**Confidence:** 3

**Summary:**

IF-Guide proactively reduces LLM toxicity by identifying and suppressing harmful training tokens using influence functions. It outperforms alignment baselines without human preference data and is computationally efficient, achieving up to 10× toxicity reduction using smaller proxy models.

**Questions:**

See Weakness.

**Ethical Concerns:**

["NO or VERY MINOR ethics concerns only"]

**Limitations:**

See Weakness.

**Paper Formatting Concerns:**

It is recommended to reduce the frequent use of the \vspace command to avoid overly tight spacing between figures and their captions, thereby improving the readability and overall layout of the manuscript.

**Quality:**

3

**Strengths And Weaknesses:**

Strengths:

1. IF-Guide leverages influence functions to proactively identify and suppress harmful tokens during training, rather than relying on post hoc alignment. This helps prevent “training-time contamination” instead of applying reactive fixes.

2. Unlike methods such as DPO and RAD, IF-Guide does not depend on costly human preference or reward data, reducing reliance on manual annotations.

Weaknesses

1. Influence function theoretically involves the inversion of the Hessian matrix, which significantly increases computational complexity. It would be helpful to provide a comparison of the computational cost of the proposed method relative to baseline approaches, to better contextualize its efficiency.

2. The current experiments are conducted on base models with fewer than 1B parameters. It remains unclear how well the proposed method generalizes to larger-scale models (e.g., 7B or 13B). Further evaluation on such models would enhance the completeness of the study.

3. Analyzing the correlation and overlap of identified toxic tokens across different base models could be an interesting direction. It may offer insights into whether different model scales attend to similar or divergent harmful patterns.

---

> ### Author Rebuttal · Authors · 2025-07-31
>
> We thank the reviewer for taking the time to read and evaluate our work. We address all the concerns and questions below and will incorporate this discussion into the final version of our paper.
>
> ---
>
> **(Weakness 1) Computational Complexity of the Inversion of the Hessian Matrix**
>
> We thank the reviewer for pointing out the complexity associated with influence functions. While influence functions traditionally involve inverting the Hessian matrix, which is indeed computationally expensive, IF-Guide avoids this bottleneck by using efficient approximations, EK-FAC [4].
>
> EK-FAC approximates the Hessian using Kronecker products, which enables efficient inversion. Prior work has demonstrated the practicality of influence functions using EK-FAC even on large models with up to 50B parameters. For example, Ruis et al. [5] show that EK-FAC computation costs are comparable to performing approximately O(100k) gradient update steps.
>
> We also note that IF-Guide is agnostic to the specific influence function approximation used. In scenarios where computational cost is a primary constraint, more lightweight alternatives such as LoGRA [6] can be adopted without modifying the overall framework.
>
> Moreover, our influence scores are computed once and can be reused across model checkpoints or even transferred across models with similar architectures (as shown in our proxy experiments). This amortization further improves the overall efficiency of our approach.
>
> We will include these clarifications and empirical cost details in the final version of the paper.
>
> [4] George et al., Fast Approximate Natural Gradient Descent in a Kronecker-factored Eigenbasis, NeurIPS (2018)
>
> [5] Ruis et al., Procedural Knowledge in Pretraining Drives Reasoning in Large Language Models, ICLR (2025)
>
> [6] Choe et al., What is Your Data Worth to GPT? LLM-Scale Data Valuation with Influence Functions, arXiv Preprint (2024)
>
> ---
>
> **(Weakness 2) Scalability of IF-Guide**
>
> We thank the reviewer for the suggestion to enhance the completeness of our evaluation. We note that full pretraining on a trillion-token dataset requires several hundred thousand GPU hours [1], which is infeasible at an academic scale. Instead, we maximized our available resources by pretraining billion-parameter models on a 1B-token dataset using 8$\times$ NVIDIA H100 GPUs, at an estimated cost of ~$10,000 USD.
>
> [1] Touvron et al., Llama 2: Open Foundation and Fine-Tuned Chat Models, arXiv Preprint (2023)
>
> Below, we provide our response along with additional evaluation conducted at this available scale.
>
> IF-Guide scales to larger models. We demonstrate the scalability in the fine-tuning setting. We fine-tune Pythia-12B—which is 10$\times$ larger than our base models—using influence scores computed on the smaller proxy model, Pythia-1B. Below, we present fluency and toxicity metrics on the RTP benchmark across fine-tuning steps:
>
> | **Dataset Size in Tokens**                                | **All Prompts** |       | **Toxic Prompts** |       | **Non-Toxic Prompts** |       | **OWT** | **LAMBADA** |
> |------------------------------------------|:-----------------:|:-------:|:--------------------:|:-------:|:------------------------:|:-------:|:---------:|:-------------:|
> |                           | EMT             | TP    | EMT                | TP    | EMT                    | TP    | PPL     | Acc.        |
> | **0 (Base Model)**        | 0.595           | 0.604 | 0.822              | 0.859 | 0.369                  | 0.345 | 9.35    | 0.739       |
> | **260M**                  | 0.120           | 0.070 | 0.158              | 0.106 | 0.081                  | 0.033 | 9.93    | 0.747       |
> | **520M**                  | 0.105           | 0.056 | 0.137              | 0.083 | 0.073                  | 0.028 | 10.01   | 0.750       |
> | **780M**                  | 0.125           | 0.076 | 0.173              | 0.121 | 0.077                  | 0.032 | 9.86    | 0.749       |
>
> At just 520M fine-tuning tokens, IF-Guide reduces EMT and TP by 5.7–10.8$\times$ with negligible fluency loss; our method is effective for large-scale models even when the proxy is $>$10$\times$ smaller.
>
> Beyond 12B parameter models, we refer to the standard influence function methods, which have been effectively scaled up to 50B parameter LLMs [2, 3]. IF-Guide adds negligible overhead to these methods, making it deployable within the same compute budgets used in these prior works. As we demonstrate, proxy models further reduce costs, further enabling scalability.
>
> [2] Grosse et al., Studying Large Language Model Generalization with Influence Functions, arXiv Preprint (2024)
>
> [3] Choe et al., What is Your Data Worth to GPT? LLM-Scale Data Valuation with Influence Functions, arXiv Preprint (2024)
>
> We will include these additional experiments and discussion in our final paper.
>
> ---
>
> **(Weakness 3) Overlap of Identified Toxic Tokens**
>
> This is a valuable suggestion, and we explore this by computing the overlap between toxic tokens identified by different models. Let $T_i$ denote the set of toxic tokens selected by model $i$. For model’s $i$ and $j$, we compute the Overlap Coefficient as follows:
>
> $$oc(i, j) = \frac{|T_i \cap T_j|}{\min(|T_i|, |T_j|)}.$$
>
> The table below reports the Overlap Coefficient values for all model pairs used in our evaluation.
>
> |                | Pythia-160M | Pythia-410M | Pythia-1B | Llama-3.2-1B |
> |---------------|:-------------:|:-------------:|:-------------:|:-------------:|
> | **Pythia-160M** | -           | 0.4661      | 0.4537    | 0.4185       |
> | **Pythia-410M** | -           | -           | 0.5168    | 0.4014       |
> | **Pythia-1B**    | -           | -           | -         | 0.4401       |
> | **Llama-3.2-1B** | -           | -           | -         | -            |
>
> The average overlap across all model pairs is 44.61% (≈8.9M out of 20M selected tokens). While this may seem modest at first glance, it is substantial when considering that the size of the training corpus is 1B tokens.
>
> Overlap is higher within the same model family: Pythia model pairs average 47.22% overlap, while cross-family pairs average 42%. Overlap is also higher for similar scales: models within $\sim$500M parameters share 47.27% of selected tokens, compared to 43.61% for models differing by $>$500M.
>
> Our results suggest that the model family is a stronger indicator of overlap than scale alone, providing a practical guideline for selecting proxy models. We will include this valuable experiment and discussion in the final version of our paper.

---

> > ### Author Response · Authors · 2025-08-05
> > **We Look Forward to Your Feedback**
> >
> > Dear Reviewer 9ufk,
> >
> > As the deadline for the discussion period is approaching in roughly three and a half days, we would like to inquire whether our response has adequately addressed the questions and concerns you raised.
> >
> > If there are any additional questions, we would greatly appreciate your feedback by tomorrow, so that we have at least 24 hours to respond and address any follow-up concerns.
> >
> > Thank you for your time, and we look forward to hearing from you.
> >
> > Sincerely,
> >
> > The Authors of Paper 25176

---

> > > ### Comment · Reviewer_9ufk · 2025-08-06
> > >
> > > Thanks for your response. My concerns have been addressed.

---

### Official Review · Reviewer_cfXp · 2025-07-03

**Clarity:** 3
**Significance:** 3
**Originality:** 3
**Rating:** 5
**Confidence:** 3

**Summary:**

This paper uses influence functions to identify which tokens in the training data is toxic and proposes methods to suppress that. Influence functions from Koh and Liang is used to identify the toxic tokens. however, they show that standard influence function based methods are not sufficient to uncover toxic tokens. They propose differential and token-level attribution to particularly choose toxic tokens ignoring the benign tokens.

**Questions:**

See the above section for questions

**Ethical Concerns:**

["NO or VERY MINOR ethics concerns only"]

**Limitations:**

I could not find a limitations section in the paper. One potential limitation might be how to scale this method for larger pretraining corpuses and larger models.

**Paper Formatting Concerns:**

No major concern

**Quality:**

3

**Strengths And Weaknesses:**

The paper presents the application of Influence Functions for identifying toxic tokens in the pretraining set. This is an important topic and paper presents a novel solution with practical considerations for applicability to 1 billion token pretraining set. The experimental section is strong with a large number of insightful experiments. The experiments show the effectiveness of the method in terms of low EMT and TP compared to the baselines. Section 4.5 also addresses the effect of proxy models for toxicity detection and application to a different model. Overall, I believe it is a strong paper.

The only concern I have is how well will this method scale to trillion token level pretraining. Although the paper outlines a method for speed-up, IF computation is typically slow especially at large scales? Also the models chosen are all <=1B tokens in size. Will this method scale for larger models? Can a smaller proxy model be used for suppressed training on large models?

---

> ### Author Rebuttal · Authors · 2025-07-31
>
> We thank the reviewer for taking the time to read and evaluate our work. We also appreciate the recognition of its strengths. Below, we address the reviewer’s concern about the scalability. We will incorporate this discussion into the final version of our paper.
>
> ---
>
> **(Weakness 1) Scalability of IF-Guide**
>
> We first emphasize that IF-Guide offers two complementary approaches, each tailored to address scalability challenges: (1) Pretraining: The method identifies toxic tokens and their linguistic context, and suppresses them during pretraining. (2) Fine-tuning: The toxic tokens identified by our method are used to guide suppression during fine-tuning. While our pretraining experiments are conducted on a 1B-token dataset, our fine-tuning evaluations are performed on models that have already been pretrained on trillion-token-scale datasets.
>
> We note that full pretraining on a trillion-token dataset requires several hundred thousand GPU hours [1], which is infeasible at an academic scale. Instead, we maximized our available resources by pretraining billion-parameter models on a 1B-token dataset using 8$\times$ NVIDIA H100 GPUs, at an estimated cost of ~$10,000 USD.
>
> Below, we provide our response along with additional evaluation conducted at this available scale.
>
> [1] Touvron et al., Llama 2: Open Foundation and Fine-Tuned Chat Models, arXiv Preprint (2023)
>
> > How well will this method scale to trillion-token level pretraining?
>
> We believe IF-Guide remains both computationally feasible and effective at larger scales:
>
> (1) *We target toxic content, not overall corpus size.* We identify and suppress a small, representative set of toxic training examples (2% of the total dataset). Even in trillion-token scales, the fraction of suppressed tokens remains small, minimizing overhead and any downstream impact on model utility.
>
> (2) *The effectiveness saturates beyond 2%.* In addition to (1), our evaluation in Appendix D.4 shows diminishing returns in toxicity reduction beyond 2% of the corpus. This implies that a small toxic subset can guide the suppression-based objective effectively, even in much larger training scales.
>
> (3) *Toxicity is likely to remain as a sparse signal at scale.* Even in the trillion-token pre-training scenarios, toxic content will represent a small fraction, which will make our approach suitable. Our method will focus on the problematic data, not attempting exhaustive coverage. As long as IF-Guide identifies representative toxic instances, the method will propagate to reduce both explicit and implicit toxicity, as demonstrated in our evaluations.
>
>
> > Although the paper outlines a method for speed-up, IF computation is typically slow, especially at large scales?
>
> We thank the reviewer for pointing out the complexity associated with influence functions. While influence functions are traditionally slow and impractical for LLMs, IF-Guide improves the scalability by using efficient approximations, EK-FAC [4].
>
> EK-FAC approximates the Hessian using Kronecker products, which enables efficient inversion. Prior work has demonstrated the practicality of influence functions using EK-FAC even on large models with up to 50B parameters. For example, Ruis et al. [5] show that EK-FAC computation costs are comparable to performing approximately O(100k) gradient update steps.
>
> We also note that IF-Guide is agnostic to the specific influence function approximation used. In scenarios where computational cost is a primary constraint, more lightweight alternatives such as LoGRA [6] can be adopted without modifying the overall framework. This includes retaining our batching and lower-precision speed-up techniques.
>
> Moreover, our influence scores are computed once and can be reused across model checkpoints or even transferred across models with similar architectures (as shown in our proxy experiments). This amortization further improves the overall efficiency of our approach.
>
> We will include these clarifications and empirical cost details in the final version of the paper.
>
> [4] George et al., Fast Approximate Natural Gradient Descent in a Kronecker-factored Eigenbasis, NeurIPS (2018)
>
> [5] Ruis et al., Procedural Knowledge in Pretraining Drives Reasoning in Large Language Models, ICLR (2025)
>
> [6] Choe et al., What is Your Data Worth to GPT? LLM-Scale Data Valuation with Influence Functions, arXiv Preprint (2024)
>
> > Will this method scale for larger models? Can a smaller proxy model be used for suppressed training on large models?
>
> IF-Guide scales to larger models. We demonstrate the scalability in the fine-tuning setting. We fine-tune Pythia-12B—which is 10$\times$ larger than our base models—using influence scores computed on the smaller proxy model, Pythia-1B. Below, we present fluency and toxicity metrics on the RTP benchmark across fine-tuning steps:
>
> | **Dataset Size in Tokens**                                | **All Prompts** |       | **Toxic Prompts** |       | **Non-Toxic Prompts** |       | **OWT** | **LAMBADA** |
> |------------------------------------------|:-----------------:|:-------:|:--------------------:|:-------:|:------------------------:|:-------:|:---------:|:-------------:|
> |                           | EMT             | TP    | EMT                | TP    | EMT                    | TP    | PPL     | Acc.        |
> | **0 (Base Model)**        | 0.595           | 0.604 | 0.822              | 0.859 | 0.369                  | 0.345 | 9.35    | 0.739       |
> | **260M**                  | 0.120           | 0.070 | 0.158              | 0.106 | 0.081                  | 0.033 | 9.93    | 0.747       |
> | **520M**                  | 0.105           | 0.056 | 0.137              | 0.083 | 0.073                  | 0.028 | 10.01   | 0.750       |
> | **780M**                  | 0.125           | 0.076 | 0.173              | 0.121 | 0.077                  | 0.032 | 9.86    | 0.749       |
>
> At just 520M fine-tuning tokens, IF-Guide reduces EMT and TP by 5.7–10.8$\times$ with negligible fluency loss; our method is effective for large-scale models even when the proxy is $>10\times$ smaller.
>
> Beyond 12B parameter models, we refer to the standard influence function methods, which have been effectively scaled up to 50B parameter LLMs [2, 3]. IF-Guide adds negligible overhead to these methods, making it deployable within the same compute budgets used in these prior works. As we demonstrate, proxy models further reduce costs, further enabling scalability.
>
> [2] Grosse et al., Studying Large Language Model Generalization with Influence Functions, arXiv Preprint (2024)
>
> [3] Choe et al., What is Your Data Worth to GPT? LLM-Scale Data Valuation with Influence Functions, arXiv Preprint (2024)
>
> ---
>
> **Location of Our Limitation Section**
>
> We discuss the potential limitations of our work in *Appendix B of the supplementary material*.

---

> > ### Author Response · Authors · 2025-08-05
> > **We Look Forward to Your Feedback**
> >
> > Dear Reviewer cfXp,
> >
> > As the deadline for the discussion period is approaching in roughly three and a half days, we would like to inquire whether our response has adequately addressed the questions and concerns you raised.
> >
> > If there are any additional questions, we would greatly appreciate your feedback by tomorrow, so that we have at least 24 hours to respond and address any follow-up concerns.
> >
> > Thank you for your time, and we look forward to hearing from you.
> >
> > Sincerely,
> >
> > The Authors of Paper 25176

---

### Official Review · Reviewer_zZWd · 2025-07-05

**Clarity:** 3
**Significance:** 2
**Originality:** 1
**Rating:** 2
**Confidence:** 4

**Summary:**

This paper proposes IF-GUIDE, a method for proactively reducing toxicity in Large Language Models (LLMs) by intervening during the training process to identify toxicity through influence functions and mitigate their effects on training.

The core idea is to use influence functions as a data attribution technique (a concept that has been studied in literature), to then identify specific training tokens that are causally responsible for toxic model outputs.

The authors introduce several modifications to standard influence functions, including differential and token-level attribution, to improve their effectiveness for this task. They then propose a novel training objective that penalizes the model for generating these identified toxic tokens. The paper presents extensive experiments across multiple models and benchmarks often outperforming standard alignment baselines.

**Questions:**

## Some references to study and include:

#### In Toxicity

The area of toxicity mitigation is vast. The authors may find it worth their time to extend the related works section.


#### In IFs for LLMs

Furthermore, there are the following relevant works in the use of IF for LLMs, some of which are close to the author's work.

*   Choe, S. K., Ahn, H., Bae, J., Zhao, K., Kang, M., Chung, Y., ... & Xing, E. P. (n.d.). *Large-Scale Training Data Attribution with Efficient Influence Functions*.
*   Li, Z., Zhao, W., Li, Y., & Sun, J. (2024). *Do Influence Functions Work on Large Language Models?*. arXiv preprint arXiv:2409.19998.
*   Pang, J., Di, N., Zhu, Z., Wei, J., Cheng, H., Qian, C., & Liu, Y. (2025). *Token Cleaning: Fine-Grained Data Selection for LLM Supervised Fine-Tuning*. arXiv preprint arXiv:2502.01968.
*   Wu, K., Pang, L., Shen, H., & Cheng, X. (2024). *Enhancing Training Data Attribution for Large Language Models with Fitting Error Consideration*. arXiv preprint arXiv:2410.01285.
*   Xia, M., Malladi, S., Gururangan, S., Arora, S., & Chen, D. (2024). *LESS: Selecting influential data for targeted instruction tuning*. arXiv preprint arXiv:2402.04333.

**Ethical Concerns:**

["NO or VERY MINOR ethics concerns only"]

**Final Justification:**

Many of the weaknesses have only been partially addressed. For example, originality comes from applying the IF method to toxicity. Some of the references I had posted were not even there in the original manuscript. Comparisons to them have been added as an afterthought. The idea novelty is in question, but in response to another reviewer, the authors are claiming substantial novelty.

---

Additional points:

1. Please move some of the examples from the last appendix into the main, if you have space.
2. The OWT perplexity numbers (Table 1) suggests that simple word filtering has substantially lower PPL than IF-Guide. This perhaps goes against my understanding of what the authors suggested in the responses to my comments. For example, PPL drops from 17.83 to 23.25 on Llama 3.1 1B. Similar on Pythia 1B.

---

Note: It is possible that I simply did not understand the work, but the perplexity numbers do seem high at 20s (See Table 1).

For pythia: the numbers are here: (are lower)
https://arxiv.org/html/2502.15796v1

For llama 3.1 someone has computed numbers here: (Even the 2 bit quantized XXS model has lower perplexity). (are much lower).
https://huggingface.co/ThomasBaruzier/Meta-Llama-3.1-8B-Instruct-GGUF

Having said that the authors did not have time to respond to this comment of mine, maybe I misunderstood something. Hence I request other reviewers and AC to not account for this in their final assessment.

**Quality:**

3

**Strengths And Weaknesses:**

### Strengths

- The paper addresses the critical issue of LLM safety. Its proactive stance, aiming to prevent the learning of harmful associations rather than suppressing them post-hoc, is a valuable and well-motivated research direction.

- The paper's experiments are its greatest strength. The authors conduct a thorough evaluation across a range of models, datasets, and settings (pre-training and fine-tuning). The results consistently show that IF-GUIDE is effective at reducing both explicit and implicit toxicity, providing strong evidence for the method's practical utility.
---

### Weakness
1. The ability of $Q_{tox}$ to identify all forms of toxicity across a pre-training dataset, say of size trillions of tokens is bottlenecked by the coverage over this sample set of 10k examples.

2. Standard influence functions are flawed for LLMs due to factors like fitting errors—is a known issue in the literature. Concurrent work, such as Wu et al. (2024), tackles this exact problem with an alternative "debias and denoise" methodology. Furthermore, the idea of token-level data cleaning for SFT has been explored by Pang et al. (2025). The present paper fails to cite, discuss, or compare against these highly relevant works, which makes it difficult to assess the relative novelty and contribution of the proposed techniques.

3. If a model is trained to suppress all toxic content, how does it recognize toxic content?
While the authors' penalty-based objective is more nuanced than simple filtering, the paper does not empirically evaluate the model's ability to handle meta-level questions about toxicity or to refuse harmful instructions.

4.  The authors argue their "suppression" approach is superior because filtering is imprecise and data-inefficient. However, this argument is less compelling when one considers that only a tiny fraction of a multi-trillion token dataset is likely to be filtered, potentially making the collateral damage negligible.

---

> ### Author Rebuttal · Authors · 2025-07-31
>
> We thank the reviewer for their time and valuable feedback. We have our answers to all the concerns and questions below, and the updates we will make to our manuscript.
>
> ---
>
> **(Weakness 1) Representativeness of the Toxic Query Set**
>
> We first clarify that a sample size of 10k is *not* a bottleneck. In Appendix D.4, we examine the effect of sample size on toxicity reduction by increasing it to 100k. Our results show that using more than 10k samples does not improve the effectiveness of our method.
>
> We also clarify that our objective is *not* to remove all toxic tokens from the training data, but to identify a representative subset of toxic tokens that allows our suppression-based objective to effectively prevent the model from learning and exhibiting toxic behaviors. **As noted as a strength in the review**, our strong empirical results across both explicit and implicit toxicity types demonstrate the effectiveness of IF-Guide using 10k sample queries.
>
> ---
>
> **(Weakness 2) Comparison to Relevant Works**
>
> We reviewed the papers cited by the reviewer during the preparation of our manuscript. While they fall under the broad umbrella of “influence functions,” they address *fundamentally different problems and therefore do not diminish the novelty of our work*. We elaborate on the key differences below and will cite them in the Background and Related Work section for completeness.
>
> > Comparison to Wu et al. (2024)
>
> (1) Wu et al. address a different problem.
>
> Wu et al. propose a technique to address fitting errors in standard influence functions. Although we introduce techniques to improve attribution quality, our focus is not on addressing fitting errors. Instead, we *leverage* influence functions as a foundation for toxicity reduction (a different application), and subsequently introduce a substantially different methodology.
>
> (2) Furthermore, our technical novelty is orthogonal.
>
> Wu et al. propose to use a standard influence function. Applying their method directly to toxicity reduction would yield results comparable to those shown in Figure 1. In contrast, IF-Guide builds *on top of* standard influence functions, independently of their exact formulation. Although we use EK-FAC in our implementation, IF-Guide is compatible with alternative influence approximations such as LoGRA [1] or Wu et al.’s algorithm. Therefore, there is no overlap in novelty.
>
> > Comparison to Pang et al. (2025)
>
> (1) From a research standpoint, our work offers clear novelty by demonstrating that a novel adaptation of influence functions can effectively address toxicity in language models.
>
> Peng et al. address a fundamentally different problem: improving model performance during supervised fine-tuning (SFT), whereas our goal is to reduce toxicity. The only commonality is the use of influence functions, but the purposes and methodologies differ significantly, making direct comparison an apples-to-oranges scenario.
>
> (2) In addition to that, there are two fundamental differences in terms of technical novelty.
>
> Peng et al. use standard influence functions, which are not designed to specifically identify toxic tokens and cannot therefore target toxicity in the training data. Using such influence functions is the same as our result in Figure 1. Also, they remove the tokens identified by standard influence functions, whereas we suppress tokens within a context window: the identified token $t_i$, as well as its immediate neighbors $t_{i-1}$ and $t_{i+1}$ (i.e., $w=1$). In Appendix D.4 (Figure 8), we show that the approach of Peng et al. (comparable to $w=0$) only modestly reduces toxicity (from 0.58 to 0.49), whereas our method with $w=1$ reduces toxicity substantially to 0.09.
>
> [1] Choe et al., *What is Your Data Worth to GPT? LLM-Scale Data Valuation with Influence Functions*, arXiv preprint (2024).
>
> ---
>
> **(Weakness 3) Recognizing/Understanding Toxic Content**
>
> > If a model is trained to suppress all toxic content, how does it recognize toxic content?
>
> Although IF-Guide penalizes toxic tokens during training, models still recognize toxic content and leverage this signal to suppress harmful outputs. In Figure 5, we isolate the activation-space vector associated with toxicity and scale its magnitude. Unlike the base model—where increasing this component increases toxicity—in our model, increasing the component reduces toxicity, while decreasing it substantially increases toxicity. This indicates that IF-Guide learns a 1D direction in activation space that the model uses to recognize and suppress toxicity.
>
> > The paper does not empirically evaluate the model's ability to refuse harmful instructions.
>
> Our evaluation demonstrates that, regardless of whether the model understands toxic content, it is unable to generate toxic outputs. Therefore, an explicit refusal mechanism is not required. Now, a possible follow-up question is: *What if an adversarial user crafts queries to elicit toxic content?* As shown in Figure 6, this scenario is less likely in our case. Compared to both pre-trained models (Base) and safety fine-tuned models (i.e., those trained with DPO), our method reduces the likelihood of generating toxic responses by at least 40%.
>
> > The paper does not empirically evaluate the model's ability to handle meta-level questions about toxicity.
>
> No prior work on reducing toxicity answers this, but we take a step forward to answer the reviewer’s question to the best of our ability by prompting our models with:
>
> > I will give you a sentence. Determine if the sentence is toxic or non-toxic based on this definition: Toxic content includes insults, hate speech, threats, or harassment. If it is toxic, respond with: Toxic. If it is not toxic, respond with: Non-toxic. Sentence: '{sentence}'
>
> To evaluate toxicity understanding, we fine-tuned two models on UltraChat-200k: the pre-trained Pythia-2.8B and its IF-Guide de-toxicified variant. We then prompted both models with 200 examples from RTP (100 toxic, 100 non-toxic) using the template above. We share the percentage of correctly classified examples below:
>
> | **Model** | **Toxic Examples (Correct %)** | **Non-Toxic Examples (Correct %)** |
> |---------------|:----------------------------:|:---------------------------------:|
> | **Base** | 87.5% | 27.0% |
> | **IF-Guide** | 76.7% | 34.0% |
>
> Both models recognize toxic content reasonably well but struggle with non-toxic cases. Given the comparable performance, IF-Guide does not appear to eliminate the model’s ability to reason about toxicity despite suppressing toxic tokens during training. We will include this experiment in the Appendix of our final paper.
>
> ---
>
> **(Weakness 4) Benefits of Suppression Over Filtering**
>
> We use suppression because it consistently achieves greater toxicity reduction than filtering. In Sections 4.2 and 4.4, IF-Guide outperforms filtering by 1.6–11$\times$ on explicit and implicit toxicity, and Appendix D.4 shows that using suppression over token filtering increases effectiveness by $\sim$2–3$\times$.
>
> Our original remark on data imprecision and inefficiency was intended to critique document-level filtering and standard influence functions, not to imply that filtering is infeasible at scale. While filtering’s collateral damage may be negligible for multi-trillion token datasets, suppression remains more effective at reducing toxicity. We will revise the manuscript to make this clearer.
>
> ---
>
> **Expanding the Related Work**
>
> We acknowledge that our related work can benefit from including more works on toxicity reduction.
>
> We also thank the reviewer for sharing the influence function references. While we already cite prior work applying influence functions to language models in our manuscript [2, 3], we note that they focus on different goals and do not address the attribution of toxic or harmful content. Nonetheless, we will ensure that these works are properly cited in the final version of our paper.
>
> [2] Choe et al., What is Your Data Worth to GPT? LLM-Scale Data Valuation with Influence Functions, arXiv Preprint (2024)
>
> [3] Xia et al., LESS: Selecting Influential Data for Targeted Instruction Tuning, ICML (2024)

---

> > ### Author Response · Authors · 2025-08-05
> > **We Look Forward to Your Feedback**
> >
> > Dear Reviewer zZWd,
> >
> > As the deadline for the discussion period is approaching in roughly three and a half days, we would like to inquire whether our response has adequately addressed the questions and concerns you raised.
> >
> > If there are any additional questions, we would greatly appreciate your feedback by tomorrow, so that we have at least 24 hours to respond and address any follow-up concerns.
> >
> > Thank you for your time, and we look forward to hearing from you.
> >
> > Sincerely,
> >
> > The Authors of Paper 25176

---

> ### Comment · Reviewer_zZWd · 2025-08-06
>
> Many thanks for your detailed responses to my queries.
>
> One query that I have still not been able to understand fully: given the related works I provided, can the authors provide a brief summary of how exactly they are going to position this work with respect to literature?

---

> ### Comment · Reviewer_zZWd · 2025-08-07
>
> Question) if a substantial percentage of the training corpus is IF detoxified, how does it affect other downstream performance? Does it have other deleterious effects?
>
>
> Q) In the response to reviewer 9ufk the authors say they can do fine-tuning for Pythia12B. At this point the model has already seen toxic tokens at pretraining. During post training, the model may not capture new toxic tokens, but how does it learn to get rid of existing toxicity? How much does toxicity reduce with regard to the base model?

---

> > ### Author Response · Authors · 2025-08-08
> >
> > Thank you for your thoughtful feedback. We are glad to provide further clarification and are happy to address any remaining questions or concerns before the end of the discussion period.
> >
> > ---
> >
> > **Positioning of Our Work**
> >
> > We appreciate your question and the opportunity to clarify how we will position our work with respect to the referenced literature. Our contribution offers a distinct perspective: using attribution to proactively steer model behavior during training. This progression can be summarized as:
> >
> > *Attribution* $\rightarrow$ *Gradient Suppression* $\rightarrow$ *Proactive LLM Safety*.
> >
> > To clarify this positioning in the final manuscript, we incorporate the following revisions:
> >
> > First, to better position our work within the literature on influence functions for LLMs, we add the following paragraph to Section 2 (Influence functions for LLMs):
> >
> > > Prior work on influence functions for LLMs falls broadly into two categories. Method-oriented approaches [1, 2, 4] focus on improving attribution accuracy or scalability (e.g., addressing fitting errors), while application-oriented approaches [3, 5] use attribution to curate training data or select examples for improving general model utility. Our work is orthogonal to both: we introduce a suppression-based training objective that uses attribution not as an end, but as a means to proactively reduce toxicity. Unlike token- or document-level filtering, which often removes only isolated instances, IF-Guide identifies toxicity-promoting contexts and modulates their gradient contributions, yielding substantially greater reductions in both explicit and implicit toxicity.
> >
> > Second, to clarify our contribution in the broader context of developing trustworthy language models, we append the following paragraph to Section 2 (Language model toxicity):
> >
> > > Most existing toxicity-mitigation methods are reactive—intervening only after toxic behaviors emerge—or proactive, but limited to coarse-grained data filtering. IF-Guide advances proactive safety by, to our knowledge, being the first to couple influence-based attribution with targeted gradient suppression during training, enabling models to avoid learning harmful associations in the first place. This unified, model-agnostic approach is effective and consistently outperforms filtering-based baselines across diverse toxicity benchmarks.
> >
> > In the final revision, we will ensure all relevant works from these areas are properly cited, while keeping our positioning clear: IF-Guide fills a previously unaddressed gap between attribution-only and filtering-only methods by transforming influence scores into a direct training signal for safety.
> >
> > [1] Choe et al., *What is Your Data Worth to GPT? LLM-Scale Data Valuation with Influence Functions*, arXiv preprint (2024).
> >
> > [2] Li et al., *Do Influence Functions Work on Large Language Models?*, arXiv preprint (2024).
> >
> > [3] Pang et al., *Token Cleaning: Fine-Grained Data Selection for LLM Supervised Fine-Tuning*, ICML (2025).
> >
> > [4] Wu et al., *Enhancing Training Data Attribution for Large Language Models with Fitting Error Consideration*, EMNLP (2024).
> >
> > [5] Xia et al. *LESS: Selecting Influential Data for Targeted Instruction Tuning*, ICML (2024).
> >
> > ---
> >
> > **Detoxifying Large Portions of the Training Corpus**
> >
> > We expect that detoxifying a large portion of the corpus would degrade model utility. As shown in Appendix D.4, selecting more than ~2% of the corpus lowers utility: increasing it to 2.5% raises PPL by 20.3% and lowers Acc. by 5%. While the exact amount that needs to be detoxified may vary by corpus, estimates that existing corpora contain up to 4% toxic content [1, 2] suggest that 2% is sufficient for realistic settings.
> >
> > Moreover, if a dataset contains a large proportion of toxic content, this can be accounted for without degrading utility by adjusting the suppression strength. For instance, in Appendix D.4, we find that reducing $\lambda$ to 0.5 still effectively reduces toxicity with less impact on utility. Thus, IF-Guide can adapt to datasets with increased toxicity.
> >
> > [1] Arnett et al., *Toxicity of the Commons: Curating Open-Source Pre-Training Data*, arXiv preprint (2024).
> >
> > [2] Prabhumoye et al., *Adding Instructions During Pretraining: Effective Way of Controlling Toxicity in Language Models*, EACL (2023).
> >
> > ---
> >
> > **Fine-Tuning With IF-Guide**
> >
> > During fine-tuning, IF-Guide penalizes gradients from toxic contexts learned during pre-training, leading the model to prefer safer alternatives. While we find that this approach is roughly 1.3–1.4$\times$ less effective than proactively pre-training, we introduce it as a lightweight alternative.
> >
> > As shown in the table in our response to Reviewer 9ufk, IF-Guide reduces the toxicity of Pythia-12B by 5.7–10.8$\times$ after fine-tuning (the first row shows the toxicity of the base model). This suggests that IF-Guide identifies toxic behaviors learned during pre-training, and that suppression-based training effectively removes them.

---

> > > ### Comment · Reviewer_zZWd · 2025-08-09
> > >
> > > Thanks this positioning is super useful.
> > >
> > > ---
> > >
> > > `increasing it to 2.5% raises PPL by 20.3% and lowers Acc. `
> > >
> > > This is very interesting and surprising. I read through Appendix D4 in considerably more detail now.
> > >
> > > A) Most pre-training gains are not data bottlenecked, we are compute bottlenecked. What the authors suggest may be true of very early pre-training, where the ppl curves are rather steep.
> > >
> > > B) Why is it that it remains stable up to 2% and suddently decreases at 2.5%? (Appendix D4 referenced by the authors). Is there noise?
> > >
> > > C) The quoted ppl numbers at 20 seems rather high, even for small models. What was this perplexity tested on? Typically a 1B size model should have a considerably lower perplexity than 20 over say a PILE dataset (or some other standard one)? Is my understanding wrong? Is this standard for Pythia model (2023)?
> > >
> > > ---
> > >
> > > My apologies for going through the Appendix D4 in detail only late. I will take into account that the authors may not have sufficient time to respond to this in my final assessment. I thank the authors for their time and patient replies.

---

### Note · Authors · 2025-08-15

Dear Reviewers, AC, and SAC,

We thank the reviewers for their time and thoughtful feedback. We are encouraged that reviewers recognized our work’s strong motivation in addressing a critical issue [zZWd, cfXp]; its novel perspective [zZWd, 9ufk] and approach [cfXp]; its thorough experiments [zZWd, cfXp]; and its demonstrated effectiveness in both pre-training and fine-tuning settings [zZWd, cfXp, AngH].

The review process has substantially strengthened our manuscript. We believe our rebuttal has addressed all concerns, resulting in a more rigorous and clear presentation.

Below, we respond to the remaining questions from Reviewer zZWd. We appreciate the opportunity to refine our work and are confident that the revised manuscript offers a sound and significant contribution to the field of trustworthy language models.

Warm regards,
The Authors

---

**Why is training only stable up to 2% toxic tokens?**

This threshold likely reflects the true proportion of toxic content in our corpus rather than noise. When too many tokens are selected, we begin suppressing a significant number of benign tokens that are important for maintaining utility. Across all our models, a 2% threshold produced consistently strong results, suggesting it is near the actual proportion of toxic content in OpenWebText.

---

**The PPL numbers seem rather high.**

We compute PPL on a holdout test set of examples from OpenWebText. The higher values are due to our academic compute budget, which limits us to training on a 1-billion-token dataset rather than the “compute-optimal” ~20-billion-token size recommended in prior work [1]. Nevertheless, these values are consistent with those reported in previous studies on toxicity reduction using comparable model and dataset sizes (e.g., GPT-2) [2, 3]. We also find that our corpus size is sufficient for generating coherent, high-quality text (see Appendix F for examples).

In our fine-tuning experiments, we start from pre-trained models trained on a much larger corpus (the Pile). These models consistently achieve lower PPL (9.9–10.6), even after IF-Guide is applied, indicating that IF-Guide scales favorably with corpus size.

[1] Hoffmann et al., *Training Compute-Optimal Large Language Models*, NeurIPS 2022.

[2] Liu et al., *DExperts: Decoding-Time Controlled Text Generation with Experts and Anti-Experts*, ACL 2021.

[3] Lee et al., *A Mechanistic Understanding of Alignment Algorithms:  A Case Study on DPO and Toxicity*, ICML 2024.

---

### Decision · Program_Chairs · 2025-09-17

**Decision:**

Accept (poster)

**Comment:**

The core idea of the paper is well motivated and effective in practice. Reviewers broadly agree the empirical story is strong with consistent reductions in both explicit and implicit toxicity across models and settings, robustness to jailbreak attacks and useful ablations. The rebuttal strengthened the submission with clearer positioning, additional analyses and a 12B fine-tuning result showing encouraging transfer without noticeable fluency loss. Overall, the work reads as an important contribution that advances proactive detoxification beyond coarse filtering or post-hoc alignment.

What would improve the paper further are a few clarifications and consolidations. The largest remaining friction points are largely presentational such as reconciling the perplexity discrepancy on the same test sets and pipelines, surfacing an explicit table that separates the one-time IF scoring cost from the training step overhead. In addition, expanding related work to precisely position novelty relative to existing IF for LLMs would also improve the impact.